# Annexin A1 Expression Is Associated with Epithelial–Mesenchymal Transition (EMT), Cell Proliferation, Prognosis, and Drug Response in Pancreatic Cancer

**DOI:** 10.3390/cells10030653

**Published:** 2021-03-15

**Authors:** Masanori Oshi, Yoshihisa Tokumaru, Swagoto Mukhopadhyay, Li Yan, Ryusei Matsuyama, Itaru Endo, Kazuaki Takabe

**Affiliations:** 1Department of Surgical Oncology, Roswell Park Comprehensive Cancer Center, Buffalo, NY 14263, USA; masa1101oshi@gmail.com (M.O.); Yoshihisa.Tokumaru@roswellpark.org (Y.T.); Swagoto.Mukhopadhyay@RoswellPark.org (S.M.); 2Department of Gastroenterological Surgery, Yokohama City University School of Medicine, Yokohama, Kanagawa 236-0004, Japan; ryusei@yokohama-cu.ac.jp (R.M.); endoit@med.yokohama-cu.ac.jp (I.E.); 3Department of Surgical Oncology, Graduate School of Medicine, Gifu University, 1-1 Yanagido, Gifu 501-1194, Japan; 4Department of Biostatistics & Bioinformatics, Roswell Park Comprehensive Cancer Center, Buffalo, NY 14263, USA; li.yan@roswellpark.org; 5Department of Gastrointestinal Tract Surgery, Fukushima Medical University School of Medicine, Fukushima 960-1295, Japan; 6Department of Surgery, Jacobs School of Medicine and Biomedical Sciences, University at Buffalo the State University of New York, Buffalo, NY 14263, USA; 7Department of Surgery, Niigata University Graduate School of Medical and Dental Sciences, Niigata 951-8510, Japan; 8Department of Breast Surgery and Oncology, Tokyo Medical University, Tokyo 160-8402, Japan

**Keywords:** ANXA1, biomarker, gene set, GSEA, metastasis, pancreatic cancer, survival, treatment response, tumor microenvironment

## Abstract

Annexin A1 (*ANXA1*) is a calcium-dependent phospholipid-binding protein overexpressed in pancreatic cancer (PC). *ANXA1* expression has been shown to take part in a wide variety of cancer biology, including carcinogenesis, cell proliferation, invasion, apoptosis, and metastasis, in addition to the initially identified anti-inflammatory effect in experimental settings. We hypothesized that *ANXA1* expression is associated with cell proliferation and survival in PC patients. To test this hypothesis, we analyzed 239 PC patients in The Cancer Genome Atlas (TCGA) and GSE57495 cohorts. *ANXA1* expression correlated with epithelial–mesenchymal transition (EMT) but weakly with angiogenesis in PC patients. *ANXA1*-high PC was significantly associated with a high fraction of fibroblasts and keratinocytes in the tumor microenvironment. *ANXA1* high PC enriched multiple malignant gene sets, including hypoxia, tumor necrosis factor (TNF)-α signaling via nuclear factor-kappa B (NF-kB), and MTORC1, as well as apoptosis, protein secretion, glycolysis, and the androgen response gene sets consistently in both cohorts. *ANXA1* expression was associated with TP53 mutation alone but associated with all KRAS, p53, E2F, and transforming growth factor (TGF)-β signaling pathways and also associated with homologous recombination deficiency in the TCGA cohort. *ANXA1* high PC was associated with a high infiltration of T-helper type 2 cells in the TME, with advanced histological grade and *MKI67* expression, as well as with a worse prognosis regardless of the grade. *ANXA1* expression correlated with a sensitivity to gemcitabine, doxorubicin, and 5-fluorouracil in PC cell lines. In conclusion, *ANXA1* expression is associated with EMT, cell proliferation, survival, and the drug response in PC.

## 1. Introduction

Pancreatic cancer (PC) ranks as the fourth most common cause of cancer-related deaths worldwide. Its five-year overall survival is less than 5% and is regarded as one of the most devastating cancer diagnoses [1]. Numerous studies over the past decades have targeted PC biology and uncovered mutations in *KRAS*, *p53*, cyclin-dependent kinase inhibitor 2A (*CDKN2A*), and *SMAD4* and their signaling pathways. These signaling pathways and the associated aberrant activation of genes play critical roles in PC progression [2]. However, the clinical relevance of these basic science findings remains vague due to a lack of studies using large patient cohorts. Recent advances in the high-volume comprehensive genomic sequencing of human tumor samples can help link the PC underlying mechanisms with clinical practice. Analyses using algorithms on comprehensive transcriptomes enable a deeper understanding of the clinical relevance of various signaling pathways and immune status within human cancers. For example, the Gene Set Variation Analysis (GSVA) allows us to understand multiple signaling pathways’ biological activity [3]. The xCell algorithm permits us to measure the fractions of 64 infiltrating cell types in the tumor microenvironment (TME) [4]. This approach has already yielded several candidates for prognostic biomarkers. Yamazaki et al. reported that epithelial–mesenchymal transition (EMT) activity in PC is a promising prognostic biomarker. Our group reported that high activity of the G2M checkpoint pathway [5] and lympho-vascular invasion [6] is associated with worse survival. In contrast, the abundance of mature blood vessels [7] and fibroblasts in PC [8] is associated with better survival. The transcriptome analysis may also uncover potential therapeutic targets for PC.

Annexin A1 (*ANXA1*, also known as lipocortin I) is a member of the annexin family of calcium-dependent phospholipid-binding proteins located on the cytosolic face of the plasma membrane and inhibits phospholipase A2 [9,10]. *ANXA1* preserves the cytoskeleton integrity and plays a significant role in the malignant phenotypes of cancer cells in vitro [11]. *ANXA1* is known to play a wide variety of functions in cancer biology, including carcinogenesis, cell proliferation, apoptosis, invasion, and metastasis, in addition to an anti-inflammatory effect [12,13]. *ANXA1* regulates transforming growth factor (TGF)-β signaling and promotes epithelial–mesenchymal transition (EMT) [14]. We previously reported that the high expression of *ANXA1* is significantly associated with inflammation, angiogenesis, and mast cell infiltration in breast cancer using in silico analyses [15]. Some suggest *ANXA1* is an attractive prognostic and predictive marker of PC due to its role in metastasis based upon in vivo experiments [11]. In addition to its relationship with cancer cells, *ANXA1* expression is also associated with multiple cells in the TME, such as fibroblasts, and, with angiogenesis, the generation of new vessels and metastasis [16,17]. Novizio et al. reported that the *ANXA1* extracellular vesicle (EV) complex participates in tumor cells–stroma intercommunication as a vehicle during PC progression, suggesting that *ANXA1* may have potential prognostic and diagnostic roles [18].

Here, we hypothesized that *ANXA1* expression is associated with cell proliferation and survival in PC and tested this hypothesis using multiple large patient cohorts.

## 2. Results

### 2.1. Annexin A1 (ANXA1) Expression Correlates with Epithelial–Mesenchymal Transition (EMT) but Not with Angiogenesis or Mature Vessel Formation in Pancreatic Cancer (PC)

Since *ANXA1* expression was linked to EMT in multiple cancer types [19,20,21], we first investigated the relationship between *ANXA1* expression and EMT in PC. The EMT pathway activity was measured using the gene set variation analysis (GSVA) algorithm, following the method we previously reported [5,22,23,24]. Concordantly, we found that *ANXA1* expression significantly correlated with the EMT pathway score in PC consistently in both The Cancer Genome Atlas (TCGA) and GSE57495 cohorts (Figure 1A; Spearman’s rank correlations (*r*) = 0.453 and 0.536, respectively; all *p* < 0.01). The low and high expression of *ANXA1* was determined by the median within each cohort (Appendix A). Further, EMT-associated genes, *CDH1* (Cadherin 1), *CDH2*, *SNAI1* (Snail Family Transcriptional Repressor 1), *SNAI2*, and *TWIST1* (twist family BHLH transcription factor 1) were all elevated in *ANXA1* high PC consistently in both cohorts, except for *CDH1* in the GSE57495 cohort. We found that other EMT-associated genes, including *FN1* (Fibronectin 1), *VIM* (Vimentin), and *TGFBI* (transforming growth factor, beta-induced), were also significantly elevated in *ANXA1* high PC in both cohorts (Appendix A). Further, we found that almost all of the expressions of genes that constitute the EMT pathway were significantly correlated with *ANXA1* expression (Appendix A). We previously published that *ANXA1* expression was associated with angiogenesis in breast cancer [15] and that the abundance of mature blood vessels was associated with better survival [7]; thus, it was of interest whether this was the case in PC. We found that *ANXA1* expression was weakly correlated with the angiogenesis score measured by the GSVA algorithm consistently in both the TCGA and GSE57495 cohorts (Figure 1C). There was no association between the *ANXA1* expression and angiogenesis-related cells such as endothelial cells, microvascular (mv), and lymphatic (ly) endothelial cells, except for mv endothelial cells in the GSE57495 cohort (Figure 1D). Generally, there was no association between *ANXA1* expression and mature blood vessel-related gene expressions, such as *PECAM1* and *S1PR1*, except for *PECAM1* in the TCGA cohort (Figure 1E). However, *ANXA1* high PC was significantly associated with a high fraction of fibroblasts and keratinocytes in both cohorts (Figure 1F). These results show that *ANXA1* expression is associated with EMT but only weakly with angiogenesis.

### 2.2. Annexin A1 (ANXA1) High PC Enriched Multiple Malignant Pathways

To better understand the functional characteristics of *ANXA1* high PC, we performed a pathway analysis using the Gene Set Enrichment Analysis (GSEA) with Hallmark gene sets in the TCGA and GSE57495 cohorts. In both cohorts, *ANXA1* high PC consistently enriched multiple malignant pathways such as hypoxia, transforming growth factor (TGF)-β signaling, TNF-α signaling via nuclear factor-kappa B (NF-kB), and MTORC1, as well as apoptosis, protein secretion, glycolysis, and the androgen response (Figure 2; all gene sets; normalized enrichment score (NES) > 1.5, false discovery rate (FDR) < 0.25). These results suggest that *ANXA1* high PC is associated with EMT and other malignant pathways in PC.

### 2.3. ANXA1 High PC Was Associated with Homologous Recombination Deficiency (HRD), TP53 Mutation, and Other Signaling Pathways but Not with Mutation Load

The mutation of *TP53*, *KRAS*, *CDKN2A*, *and SMAD4* plays a key role in the carcinogenesis of PC. As such, it was of interest whether *ANXA1* expression related to the mutation load and HRD, as well as the mutation rates and signaling pathways of these genes in the TCGA cohort. We found that *ANXA1* high PC was not associated with either silent or non-silent mutation rates, fraction altered, single nucleotide variation (SNV), or Indel neoantigens; however, it was associated with HRD (Figure 3A; *p* = 0.026). *ANXA1* high PC was significantly associated with high mutation rates of the *TP53* gene but not with *KRAS*, *CDK2A*, or *SMAD4* (Figure 3B). However, p53, KRAS, E2F, and TGF-β signaling pathways, measured by the GSVA algorithm, were all significantly associated with *ANXA1* expression (Figure 3C, all *p* < 0.001). These findings suggest that *ANXA1* high PC are associated with HRD; TP53 mutation; and the signaling of *TP53*, *KRAS*, *CDKN2A*, and *SMAD4*.

### 2.4. ANXA1 High PC Was Associated with High Infiltration of T-Helper Type 2 (Th2) Cells

We reported that *ANXA1* high breast cancer is associated with a high infiltration of mast cells [15]. Therefore, we wanted to investigate whether similar immune cell infiltration is seen in *ANXA1* high PC. The xCell algorithm was used to estimate the fraction of immune cells in the tumor microenvironment of PC in the TCGA and GSE57495 cohorts. We found that *ANXA1* high PC was significantly associated with a high infiltration of Th2 cells consistently in both the TCGA and GSE57495 cohorts (Figure 4; *p* = 0.003 and *p* < 0.001, respectively). Interestingly, *ANXA1* high PC was associated with a high infiltration of mast cells in the TCGA; however, it was associated with a low infiltration in the GSE57495 cohort. Furthermore, *ANXA1* high PC was associated with a low infiltration of CD8^+^ T cells and Th1 cells in the GSE57495 cohort but not in the TCGA. These findings suggest that a high expression of *ANXA1* is associated with a high infiltration of Th2 cells in PC.

### 2.5. ANXA1 High PC Is Associated with Advanced Histological Grade and with Increased Cell Proliferation

Given that *ANXA1* high PC was associated with multiple malignant pathways, including EMT, which, in turn, was related to metastasis, we expected *ANXA1* high PC to be associated with the clinical parameters of aggressiveness in PC. Although *ANXA1* expression was not associated with the American Joint Committee on Cancer (AJCC) Stage and lymph node metastasis (N-category) (Figure 5A; *p* = 0.73 and 0.582, respectively), *ANXA1* high PC was significantly associated with the advanced histologic grade (Figure 5A; *p* < 0.001). We also investigated the association of the *ANXA1* expression with the clinical features (age at diagnosis, race, AJCC T- and M-categories, primary tumor site, and histological diagnosis) in the TCGA cohort. There was no significant difference between *ANXA1* low and *ANXA1* high PC (Appendix A). On the other hand, *ANXA1* high PC was significantly associated with a high proliferation score and a high expression of *MKI67* (Figure 5B; both *p* < 0.001). Furthermore, *ANXA1* expression was significantly correlated with cell proliferation-related pathways: E2F targets, the G2M checkpoint, and Mitotic spindle, which were calculated by the GSVA algorithm (Figure 5C; E2F targets; Spearman’s rank correlation (*r*) = 0.457 and 0.392, G2M checkpoint; *r* = 0.502 and 0.391, Mitotic spindle; *r* = 0.592 and 0.452, respectively, all *p* < 0.01). These findings suggest that *ANXA1* expression significantly correlates with aggressive cell proliferation in PC.

### 2.6. ANXA1 High PC Are Significantly Associated with Worse Survival

Given that *ANXA1* expression is significantly correlated with cancer cell proliferation, we predicted that *ANXA1* high PC is associated with worse survival. To assess this, we analyzed the disease-free survival (DFS), disease-specific survival (DSS), and progression-free survival (PFS) in the TCGA cohort, as well as the overall survival (OS) in the TCGA and GSE57495 cohorts. We found that *ANXA1* high PC was significantly associated with worse DFS, DSS, and PFS in the TCGA, as well as worse OS in both cohorts (Figure 6A; all *p* < 0.05). We also analyzed whether the survival risk is the same by histological grade, since the *ANXA1* expression was higher in the advanced histological grade. We found that *ANXA1* high PC was significantly associated with a worse PFS regardless of grade 1, grade 2, or grade 3 (Figure 6; *p* = 0.040 and 0.018, respectively). These results indicate that, prognostically, an *ANXA1* high expression is significantly associated with worse survival in PC patients regardless of histological grade.

### 2.7. ANXA1 Expression Correlates with Drug Sensitivity in PC Cells

Finally, we investigated the association of *ANXA1* expression and drug response in PC cells. The list of cells used for analyses is shown in Appendix A. We found that *ANXA1* expression was significantly correlated with the area under the curve (AUC) of gemcitabine and doxorubicin in the primary PC cell lines (Figure 7; Spearman’s rank correlate coefficient^®^ = 0.587 (*p* = 0.03) and *r* = 0.535 (*p* = 0.03), respectively) and negatively correlated with the AUC of 5-fluorouracil in the metastatic PC cell lines (Figure 7; *r* = −0.553 (*p* = 0.01)). These results implicate that *ANXA1* expression may be associated with drug responses in PC that warrant further study.

## 3. Discussion

Recent advances in sequencing technology and genetic analyses allow us to investigate the functions of cancer from the transcriptome of a bulk tumor. Analyses using GSVA scores allow investigators to explore the biologic activity of varying signaling pathways and help identify the mechanisms involved. This approach is well-accepted in the field, evident from numerous citations of the original paper. Our group previously reported that the angiogenesis score was significantly associated with specific gene expressions such as VEGF-related genes, endothelial cell marker genes, and vascular-stability-related genes using this approach [24]. Cell proliferation-related pathways such as E2F targets [23], G2M checkpoint [25], and MYC target [26] scores were significantly associated with the pathological grade in breast cancer. The clinical relevance of angiogenesis [24], inflammation [27], cell proliferation-related pathways [5,23,25,28], KRAS signaling [29], and estrogen response pathways [22] have been published using the same approach. Furthermore, the xCell algorithm allows us to measure several types of cells, including immune cells and stromal cells, in the tumor microenvironment (TME). Our group previously reported the clinical relevance of CD8^+^ T cells [30], regulatory T cells [31], and dendritic cells (DC) [32], as well as fibroblasts [8], in multiple types of cancer using the xCell algorithm. The link between *ANXA1* expression and several signaling pathways and several cells in TME was elucidated using these algorithms.

Specifically, in this study, we investigated the clinical relevance of *ANXA1* expression in pancreatic cancer (PC). *ANXA1* expression correlated with EMT and its related gene expressions but very weakly with angiogenesis and had no relationship with vascular cells or mature blood vessels in PC. *ANXA1* high PC was significantly associated with a high fraction of fibroblasts and keratinocytes in the tumor microenvironment (TME). *ANXA1* high PC enriched multiple malignant pathways, including hypoxia, TNF-α signaling via NF-kB, and MTORC1, as well as the apoptosis, protein secretion, glycolysis, and androgen response gene sets by GSEA. *ANXA1* expression was associated with TP53 mutation alone but was associated with all the KRAS, p53, E2F, and TGF-β signaling pathways. *ANXA1* high PC was associated with homologous recombination deficiency but not with mutation load.

Additionally, *ANXA1* high PC was associated with high infiltration of T-helper type 2 cells in TME. Furthermore, *ANXA1* high PC was associated with the advanced histologic grade, cell proliferation, and *MKI67* expression. From a prognostic standpoint, *ANXA1* high PC was significantly associated with worse patient survival regardless of the grade. Finally, high *ANXA1* expression correlated positively with the sensitivity to gemcitabine and doxorubicin and negatively with the sensitivity to 5-fluorouracil in PC cell lines.

Although numerous publications demonstrate that *ANXA1* plays multifaceted roles in cancer development and progression, its expression and function appear to be “cancer type-specific” [33]. *ANXA1* activation was reported to play a critical role in the EMT pathway in several types of cancer, including PC, consistent with our results. Interestingly, *CDH1*, an epithelial marker that was shown to decrease with EMT, was elevated in *ANXA1* high PC in this current study, which coincides with the previous report that *CDH1* correlates with EMT [34]. Since almost all the genes in the EMT pathway gene set significantly correlated with *ANXA1* expression, we concluded the association. *ANXA1* protein could regulate metastasis by favoring cell migration/invasion intracellularly, as a cytoskeleton remodeling factor, and extracellularly like the formyl peptide receptor (FPR) ligand [11,35]. We found that not only EMT but also other signaling pathways such as protein secretion, glycolysis, and the androgen response were enriched in *ANXA1* high PC. Although *ANXA1* was initially found to inhibit the inflammatory response, that has not been shown in cancer models [20]. However, our group previously reported that *ANXA1* high breast cancer was significantly associated with inflammation and angiogenesis signaling pathways [15]. The fact that the association with inflammation or angiogenesis was not observed in PC indicates that the relationship between *ANXA1* expression and inflammation or angiogenesis may differ by the cancer type.

The gold standard to analyze the tumor microenvironment (TME), which plays a crucial role in cancer progression and treatment response, is flow cytometry or immunohistochemistry. Although these approaches are well-established, they are expensive and labor-intensive, particularly when analyzing large patient cohorts. By comparison, bioinformatic approaches can estimate the quantity and function of cells in tens of thousands of samples with less cost and time [4,36,37]. We previously reported the clinical relevance of immune cells, including CD8^+^ T cells [30], regulatory T cells [31], and dendritic cells (DC) [32], as well as stromal cells such as fibroblasts [8], in the TME using the xCell algorithm, which allows us to estimate the fraction of 64 cells in the TME with the transcriptome of a bulk tumor. Cancer of the breast, colorectal, lung, and kidney with a low expression of *ANXA1* is scarcely infiltrated by DC and cytotoxic T lymphocytes, supporting the idea that *ANXA1* deficiency facilitates immune escape [38]. The association between *ANXA1* and the tumor immune microenvironment in PC has not been fully studied. In this study, *ANXA1* high PC was found to be associated with a high infiltration of Th2 cells. Since Th2 cells are known as pro-cancer immune cells [39], high Th2 cell infiltration may explain the association between *ANXA1* high and poor prognosis in PC patients. We previously reported that *ANXA1* high breast cancer was significantly associated with high infiltration of mast cells [15]. Since the functions of *ANXA1* and mast cells vary in different cancer types, the association between *ANXA1* expression and mast cells may differ between breast cancer and PC. Several studies, including one using *ANXA1* knockout mice, showed that stroma-derived *ANXA1* expression promotes tumor growth, angiogenesis, and metastasis [16,17]. The relationship between *ANXA1* and fibroblasts that we showed is in line with that report. It is possible that *ANXA1* generated by fibroblasts contributes to its expression in bulk tumors and is one of the reasons why *ANXA1* expression was highly associated with poor prognosis.

Drug resistance is one of the major obstacles that contribute to PC mortality in clinics. Whether it is de novo or acquired, drug resistance involves numerous genetic and epigenetic alterations in PC. Numerous studies have attempted to identify the mechanisms and molecular markers involved in either de novo or acquired drug resistance processes [40,41]. Zhang et al. reported that the overexpression of *ANXA1* induced by low-concentration arsenic trioxide (ATO), an antitumor agent, makes cancer cells more resistant to the agent via activated ERK MAPKs [42]. Belvedere et al. reported that *ANXA1* expression maintains an overall aggressive phenotype and chemotherapy resistance using *ANXA1* knockout MIA PaCa-2 cells [11]. We showed results similar to this report with human patient data. We also showed the association of *ANXA1* expression with proliferation-related factors, such as *MKI67* expression, as well as gene sets of the E2F targets, mitotic spindle, and G2M checkpoints. This finding is in the setting of previous reports that high G2M checkpoint activity is significantly associated with a worse prognosis in PC [5].

Interestingly, we found that *ANXA1* is positively correlated to the apoptosis pathway, where evading apoptosis is one of the original hallmarks of cancer [43]. We believe this is because we analyzed the involvement of *ANXA1* in cancer progression—the worsening of existing cancer. This is different from the hallmarks of cancer, which are mechanisms in carcinogenesis, the development of cancer from normal cells. Further, our findings suggest that *ANXA1* high PC is associated with overwhelming cell proliferation that overrides elevated apoptosis, which may be, in part, due to activation of the p53 pathway. In the current study, we demonstrated that *ANXA1* expression is associated with poor survival, which coincides with Shang et al., who analyzed 39 pancreatic ductal adenocarcinoma patients [44], where we analyzed 239 PC patients. These results suggest that future studies investigating the relationship between *ANXA1* expression and drug response using large clinical cohorts are warranted.

We successfully demonstrated the clinical relevance of *ANXA1* in PC; however, this study still had certain limitations. This study was prone to selection bias, given it was a retrospective design using previously published cohorts. Additionally, we were able to use these cohorts to evaluate the gene expression at a single time point, at the time of surgical removal of PC, but not evaluate the change in gene expression and signaling pathways in these tumors over time. The relationship between *ANXA1* expression and the drug response was assessed using data from the cell line encyclopedia, since we did not have access to any PC patient cohorts with comprehensive transcriptomes that were associated with clinical drug response data. A prospective study will be required to prove the utility of *ANXA1* expression as a prognostic and predictive biomarker. In particular, *ANXA1* expression is expected to be a useful tool to identify which chemotherapy may respond to which patient and when by obtaining tumor samples longitudinally, including metastatic tumors.

## 4. Materials and Methods

### 4.1. Clinical and Transcriptomic Data of TCGA and GEO Cohorts in Pancreatic Cancer Patients

Transcriptomic data and mutation data of The Cancer Genome Atlas of pancreatic adenocarcinoma (TCGA-PAAD; *n* = 176) were obtained through the cBio Cancer Genomic Portal [45]. Survival data of pancreatic cancer patients were obtained from the Pan-Cancer Clinical Data Resource [46]. Clinical and transcriptomic data of the GSE57495 cohort studied by Chen et al. (*n* = 63) [47] were obtained through the Gene Expression Omnibus (GEO) repository. Log_2_-transformed gene expression data were used for all analyses.

### 4.2. Drug Sensitivity and Transcriptomic Data of Pancreatic Cancer Cell Lines

Thirty-five pancreatic cancer cell lines with both comprehensive gene expression and drug response (area under the curve (AUC)) data from the cancer cell line encyclopedia (CCLE) [48] through the Depmap portal (https://depmap.org/portal/, accessed on 1 February 2021) were used to assess the correlation between *ANXA1* expression and the drug response. The AUCs were adjusted for the range of tested drug concentrations that allowed the integration of heterogeneous drug sensitivity data from the CCLE, the Genomics of Drug sensitivity in cancer (GDSC), and the Cancer Therapeutics Response Portal (CTRP) [49]. The list of cell lines is shown in Appendix A.

### 4.3. Scores

We used several scores in this study. Homologous recombinant deficiency and mutation-related, including altered fraction, single nucleotide variant (SNV) and indel neoantigens, and silent and non-silent mutations, were calculated by Thorsson et al. in the TCGA cohort [50]. Gene set scores, including angiogenesis, E2F, G2M, and EMT, were calculated by the gene set variation analysis (GSVA) with hallmark gene sets of the Molecular Signatures Database (MSigDB) collection, as we previously reported [5,23,24,51]. We used the xCell score as the infiltrating fraction of the immune and stromal cells in the tumor microenvironment, which was calculated by the xCell algorithm [4], as we previously reported [52,53,54,55].

### 4.4. Statistical Analysis

R software (v 4.0.1, R project for Statistical Computing) and Microsoft Excel (v 16 for Windows, Redmond, WA, USA) were to analyze and generate the graphs in the study. We divided them into *ANXA1* low and *ANXA1* high groups by the median cut-off within each cohort. *p*-values were calculated by Fisher’s exact test. The Kruskal–Wallis and Mann–Whitney *U* test were used for group comparisons, as described in their respective figure legends. The Log-rank test was used for the survival analysis. *p*-values < 0.05 were used to determine the statistical significance.

## 5. Conclusions

Annexin A1 (*ANXA1*) expression is associated with EMT; multiple malignant pathways; and the infiltration of fibroblasts, keratinocytes, and T-helper type 2 cells in the tumor microenvironment. Furthermore, *ANXA1* high PC expression is associated with cell proliferation and worse patient survival and drug response.

## Figures and Tables

**Figure 1 cells-10-00653-f001:**
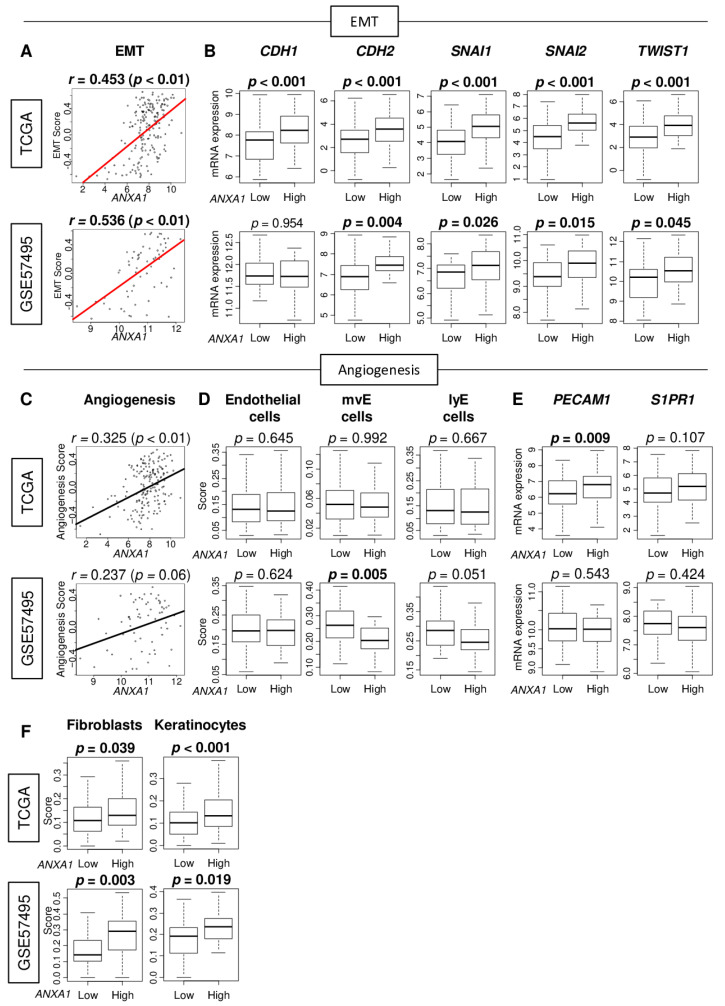
Association of Annexin A1 (*ANXA1*) expression with epithelial–mesenchymal transition (EMT), angiogenesis, and stromal cells in The Cancer Genome Atlas (TCGA) and GSE57495 cohorts. (**A**) Scatter plots of *ANXA1* expression with the EMT score. (**B**) Boxplots of EMT-associated gene expression: cadherin1 (*CDH1*) and *CDH2*, snail family transcriptional repressor 1 (*SNAI1*) and *SNAI2*, and twist-related protein 1 (*TWIST1*) by *ANXA1* low and *ALXA1* high pancreatic cancer (PC). (**C**) Scatter plots of *ANXA1* expression with the angiogenesis score. (**D**) Boxplots of infiltrating fraction of endothelial cells, microvascular endothelial (mvE) cells, and lymphatic endothelial (lyE) cells by *ANXA1* low and *ANXA1* high PC. (**E**) Boxplots of angiogenesis-associated genes expression: platelet and endothelial cell adhesion molecule 1 (*PECAM1*) *and* sphingosine-1-phosphate receptor 1 (*S1PR1*) by *ANXA1* low and *ANXA1* high PC. (**F**) Boxplots of the infiltrating fraction of fibroblasts and keratinocytes by *ANXA1* low and *ANXA1* high PC. Median cut-off within each cohort was used to divide them into *ANXA1* low and *ANXA1* high groups (*n* = 88, respectively, in the TCGA and *n* = 31 and 32, respectively, in the GSE57495 cohort). Spearman’s rank correlation was used for the correlation analysis. For group comparison, *p*-values were calculated by the Mann–Whitney *U* test.

**Figure 2 cells-10-00653-f002:**
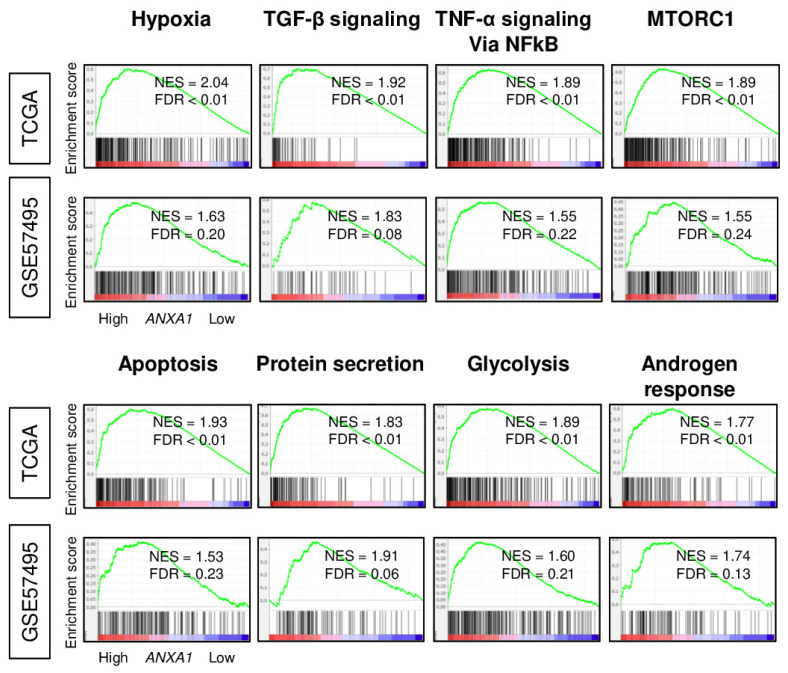
Gene set enrichment analysis of pancreatic cancer with high annexin A1 (*ANXA1*) expression in the TCGA and GSE57495 cohorts. Enrichment plots of hypoxia, transforming growth factor (TGF)-β signaling, tumor necrosis factor (TNF)-α signaling, MTORC1, apoptosis, protein secretion, glycolysis, and the androgen response of hallmark gene sets with NES and FDR. Median cut-off within each cohort was used to divide into *ANXA1* low and *ANXA1* high groups (*n* = 88, respectively, in the TCGA and *n* = 31 and 32, respectively, in the GSE57495 cohort). NES, normalized enrichment score and FDR, false discovery rate.

**Figure 3 cells-10-00653-f003:**
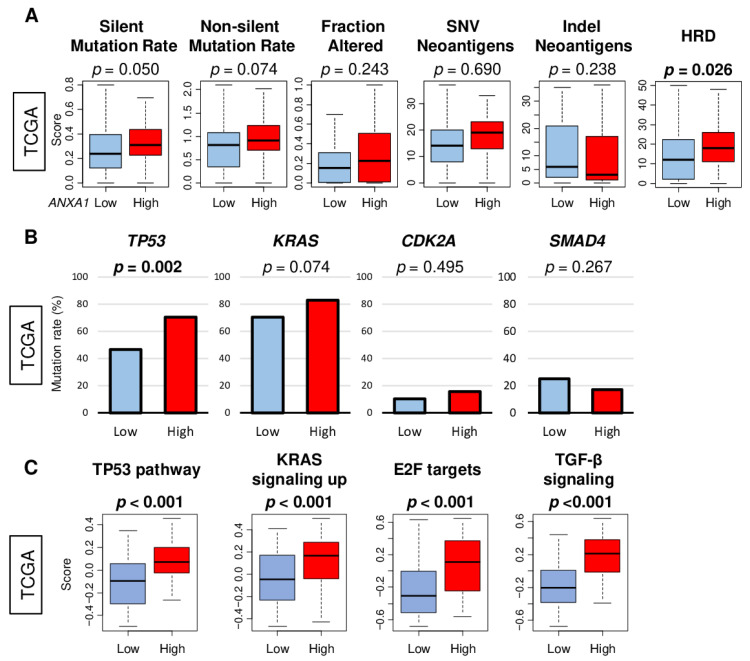
The association of *ANXA1* with gene mutation and homologous recombination deficiency (HRD) in the TCGA pancreatic cancer cohort. (**A**) Bar plots of the mutation rate of *KRAS*, *TP53*, *CDK2A*, and *SMAD4* by *ANXA1* low and *ANXA1* high groups. *p*-values were calculated with Fisher’s exact test. (**B**) Boxplots of the mutation-related scores, including altered fraction, single nucleotide variant (SNV) and indel neoantigens, and silent and non-silent mutations, and (**C**) HRD scores by *ANXA1* low (blue) and *ANXA1* high (red) groups. Median cut-off was used to divide into *ANXA1* low and *ANXA1* high groups (*n* = 88, respectively). *p*-values were calculated by the Mann–Whitney *U* test.

**Figure 4 cells-10-00653-f004:**
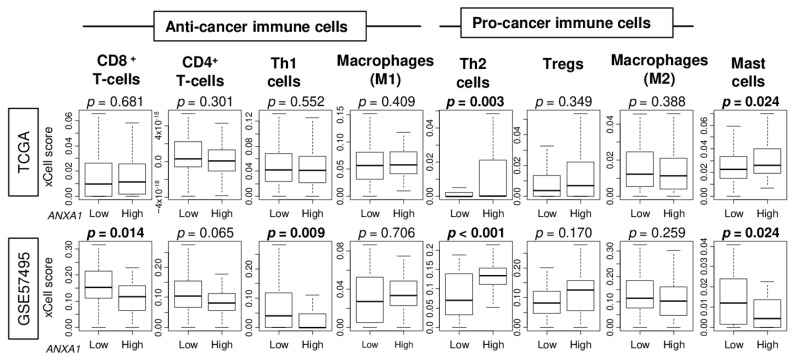
Association of the *ANXA1* expression and infiltrating immune cells in pancreatic cancer in the TCGA and GSE57495 cohorts. Boxplots of the infiltrating fraction of CD8^+^ T cells, CD4^+^ T cells, T-helper types 1 and 2 (Th1 and Th2), regulatory T cells (Tregs), M1 and M2 macrophages, and Mast cells by *ANXA1* low and *ANXA1* high groups. Median cut-off within each cohort was used to divide into *ANXA1* low and *ANXA1* high groups (*n* = 88, respectively, in the TCGA and *n* = 31 and 32, respectively, in the GSE57495 cohort). *p*-values were calculated by the Mann–Whitney *U* test.

**Figure 5 cells-10-00653-f005:**
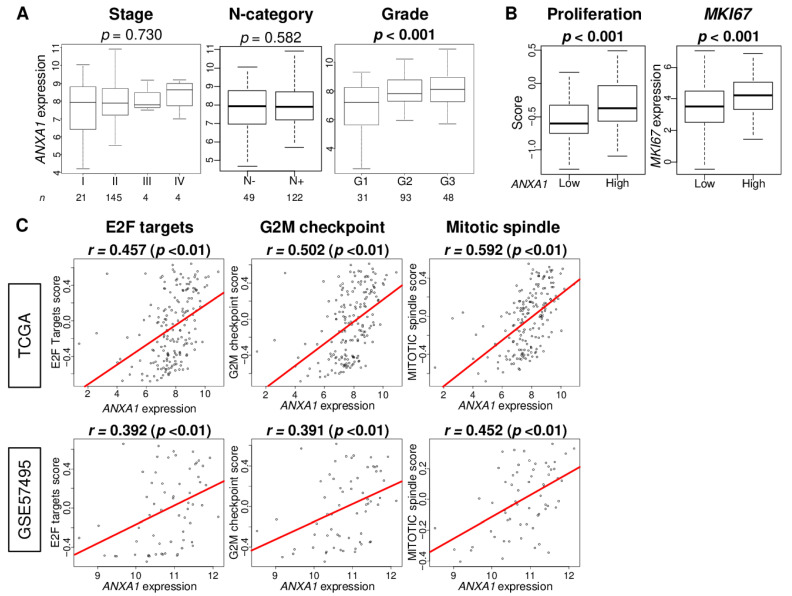
Association of the *ANXA1* expression and tumor aggressiveness in PC. Boxplots of (**A**) the clinical factors: AJCC stage, N-category, and histological grade and (**B**) proliferation-related factors: proliferation score and *MKI67* expression in the TCGA cohort. *p*-values were calculated by the Kruskal–Wallis and Mann–Whitney *U* tests. (**C**) Correlation plots between the *ANXA1* expression and cell proliferation-related score: E2F targets, G2M checkpoint, and Mitotic spindle score in the TCGA and GSE57495 cohorts. The median cut-off was used to divide them into the *ANXA1* low and *ANXA1* high groups (*n* = 88, respectively). Spearman’s rank correlation was used for the analysis. AJCC: American Joint Committee on Cancer.

**Figure 6 cells-10-00653-f006:**
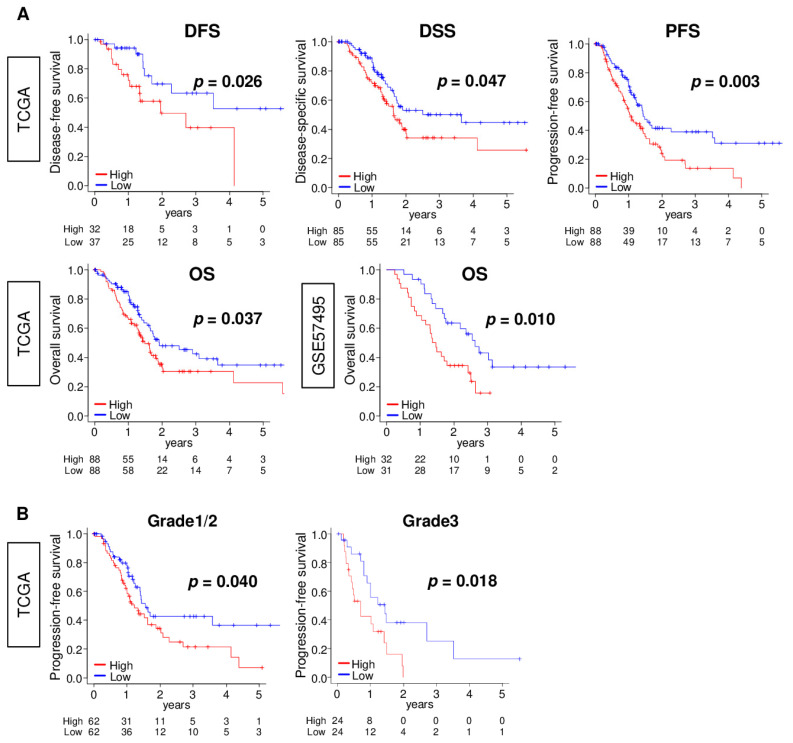
Association of the *ANXA1* expression with patient survival in the TCGA and GSE57495 cohorts. (**A**) Kaplan–Meier curve between low (blue line) and high (red line) with disease-free survival (DFS), disease-specific survival (DSS), progression-free survival (PFS), and overall survival (OS). The median cut-off within each cohort was used to divide them into the *ANXA1* low and *ANXA1* high groups (*n* = 88, respectively, in the TCGA and *n* = 31 and 32, respectively, in the GSE57495 cohort). (**B**) Kaplan–Meier curve between low and high with PFS in the pathological grades 1/2 (*n* = 62, respectively) and grade 3 groups (*n* = 24, respectively) in the TCGA cohort. *p*-values were calculated by log-rank test.

**Figure 7 cells-10-00653-f007:**
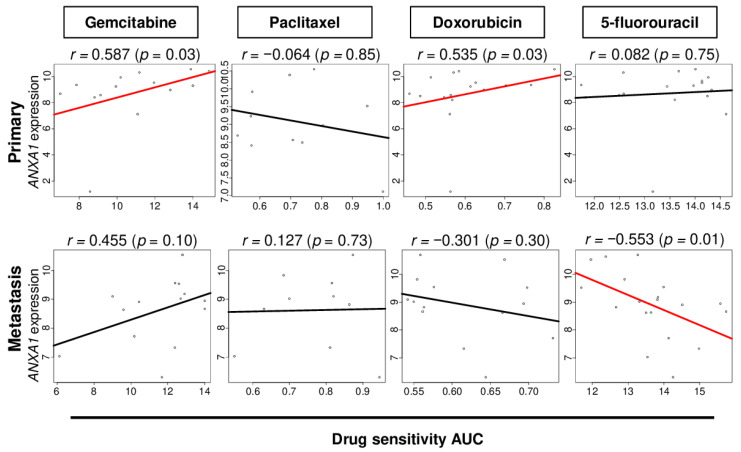
Correlation of the *ANXA1* expression with the drug sensitivity of several drugs in primary and metastasis PC cell lines. The correlation plots of the *ANXA1* expression with the level of drug sensitivity area under the curve (AUC) of gemcitabine, paclitaxel, doxorubicin, and 5-fluorouracil. Spearman’s rank correlation coefficient was used for the analysis.

## Data Availability

All data were from previous studies.

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
