# Peer review of "Annexin A1 Expression Is Associated with Epithelial–Mesenchymal Transition (EMT), Cell Proliferation, Prognosis, and Drug Response in Pancreatic Cancer"

_cells, 2021, doi:10.3390/cells10030653_

Round 1

Reviewer 1 Report

In this article, ' Annexin A1 expression is associated with epithelial-mesenchymal transition(EMT), cell proliferation, prognosis and drug response in pancreatic cancer.' authors have elucidated the role of Annexin A1 (ANXA1) in pancreatic cancer using publicly available datasets. The authors have used various analytical tools and scoring methods like GSVA, xCell to conclude that ANXA1 is associated with EMT and other cancer-related pathways. Though this study offers novel insights into the role of ANXA1, some queries need to be addressed.
1. The authors should thoroughly check the article for its sentence formation and grammar.
2. Authors should show the levels of ANXA1 in PC cases from both cohorts.
3. It is not clear how authors have concluded that ANXA1 expression is correlated with EMT, as levels of CDH1 are high in PC samples with high ANXA1 levels. CDH1 is an epithelial marker, and with EMT, its expression decreases. Have authors checked other genes' expression (apart from those discussed in the article) that are part of the EMT pathway?
4. A decrease in apoptosis is the hallmark of cancer. However, fig 2 shows that ANXA1 is positively correlated to the apoptosis pathway. Can the authors explain this finding
5. Also, similar findings are reported for the G2/M checkpoint and TP53 pathway. As both pathways block cancer, can the authors justify the positive association of ANXA1 with these pathways?
6. There is not enough information on the cell lines used (source, culture conditions), doses, and time points for drug treatment.
7. Also, it is difficult to conclude anything about ANXA1 as a predictive biomarker for the drugs used.

Reviewer 2 Report

The work by Oshi et al. appears well done and can be considered as an interesting new issue regarding the PC characterization, mainly focused on the role of the protein ANXA1.

Therefore, there are some concerns which required to be revised. Thus, in my opinion, the paper has to be subjected to a minor revision, just considering that it does not need substantial experimental changes but important modification have to be done about the text.

First, the paper is an overall bioinformatic extrapolation of data yet included in preexisting datasets. Thus, I suggest to the authors to underline in the last part of the paper the kind of practical experiments they thought to perform in the future based on what they have chosen to publish here.

The data from which they start the analysis are reported in dataset derived from clinical cohorts studies. Moreover, the results they have obtained are not well described. For examples, the correlation found about the ANXA1 expression  and the angiogenesis, in which way they can affirm it? The angiogenesis has been previously proved by specific genes expression or through the presence of capillary-like structures in the tissues highlighted by staining techniques? The same consideration can be applied to all the other kinds of correlation: how the authors define the immune cells and the fibroblasts presence? Please, described in depth the sample characterization at the base of the correlation studies.

Additionally, the authors made a correlation between ANXA1 expression and drug sensitivity of PC cells. As for the previous point, I encourage a description of the studies at the base of this concept in order to better define the finding here reported. Furthermore, what about the drug response in patients? There is the possibility to include this kind of data in order to enrich the information about the correlation with ANXA1 expression?

The suggestion the authors reported in the Discussion section about the role of fibroblasts deriving microvesicles and their influence on PC progression, correlated with ANXA1 expression, is an interesting point of view which deserves to be deepened. Therefore, in the literature there are many scientific evidence regarding the role of PC microvesicles on the TME in autocrine and paracrine manner, as for the PMID: 33353163 which I suggest to report at least in the Introduction in order to give an more complete vision.

Lastly, what about the bold character in the Introduction and Discussion sections?  Please, also revised the English language and several grammatical errors.

Reviewer 3 Report

Major Points:

The authors should indicate what percentage of the TCGA and GSE57495 data sets are ANXA1 high as defined in this study. It is important to highlight what proportion of PC patients will benefit from the findings of this study. [Also refer to study PMID: 15133855]

It is unclear what the number of ANXA1 high and low cases are in the GSE cohort (176 TCGA, ANXA1 high n=88) . If n is different for each analysis please note in the figure legends. If it is the same for all figures then please note in the methods section (in addition to the note in supplementary).  

The manuscript has many typographical errors. Please fix them.

Log-lank---Log-rank

Spherman---Spearman

Valiation---Validation

Commas ----decimal points

Minor Point:

It may be better to consistently use ANXA1 high or ANXA1 low instead of “ANXA1 expression high XXX cancer” used in many places. Makes the text very unclear and difficult to follow.

Round 2

Reviewer 1 Report

The authors have addressed most of the major concerns/ queries.

There are a couple of minor concerns that need attention.

  1. The authors haven't performed any experiments; instead, they have used cell line data to understand the relationship between ANXA1 and drug response. Hence, line 385 should be 'The relationship between ANXA1 expression and drug response was assessed using data from cell line encyclopedia rather than 'The relationship between ANXA1 expression and drug response was assessed only in vitro.'
  2. It is also difficult to understand the observation on the causality of ANXA1 expression (line number 399-402). If this statement is not adding enough value to an overall conclusion, it can be omitted. 

Reviewer 2 Report

Based on the modification perfomed by the authors answering to the my opinion, I consider this paper acceptable for publication.

Reviewer 3 Report

There are still minor English language/sentence errors in the text. Please take a thorough and final look.